# Structural Characteristics and Evolutionary Drivers of Global Virtual Water Trade Networks: A Stochastic Actor-Oriented Model for 2000–2015

**DOI:** 10.3390/ijerph20043234

**Published:** 2023-02-12

**Authors:** Lizhi Xing, Wen Chen

**Affiliations:** 1College of Economic and Management, Beijing University of Technology, Beijing 100124, China; 2International Business School, Beijing Foreign Studies University, Beijing 100089, China

**Keywords:** virtual water trade, stochastic actor-oriented model, multi-regional input–output table, ecologically unequal exchange, water endowment

## Abstract

The globalization of trade has caused tremendous pressure on water resources globally, and a virtual water trade provides a new perspective on global freshwater sharing and water sustainability. No study has yet explored the structural characteristics and drivers of the evolution of global virtual water trade networks from a network structure evolution perspective. This paper aims to fill this critical gap by developing a research framework to explore how endogenous network structures and external factors have influenced the evolution of virtual water trade networks. We constructed virtual water trade networks for 62 countries worldwide from 2000 to 2015 and used an innovative combination of multi-regional input–output data and stochastic actor-oriented models for analytical purposes. Our results support the theoretical hypothesis of ecologically unequal exchange and trade drivers, arguing that virtual water flows from less developed countries to developed countries under global free trade and that unequal trade patterns lead to excessive consumption of virtual water in less developed countries. The results partially support the theoretical content of water endowment and traditional gravity models, finding that trade networks are expanding to farther and larger markets, confirming that national water scarcity levels do not impact the evolution of virtual water trade networks. Finally, we point out that meritocratic links, path dependence, reciprocity, and transmissive links have extreme explanatory power for the evolutionary development of virtual water networks.

## 1. Introduction

With the development of economies and the increase in population, the global demand for water resources continues to increase, and human beings make unreasonable developments of water resources. As a result, many countries and regions are experiencing water shortages of varying degrees, which has become a major challenge facing the world in the 21st century. Studies have shown that water scarcity and water pollution problems will continue to intensify in the coming decades, thus causing a series of problems such as limiting economic development, lack of environmental sustainability, and food insecurity [1,2,3]. In 1993, Tony Allan introduced the concept of virtual water, which is the amount of water needed to produce products and services or the virtual water embedded in the product and service volume [4]. Since then, virtual water has been widely used as a quantitative evaluation indicator of the “water” production chain for resource inputs, commodity trade, and economic output, and has played an important role in exploring the mechanisms of water flows in the context of economic trade [5,6]. This flow of water resources with global trade is called virtual water, which refers to water trade when goods and services are exchanged virtually rather than in the real sense [7]. It provides a new perspective on global freshwater sharing and water sustainability [8].

In order to achieve water savings in global commodity exchange, Tony Allan first proposed a “virtual water strategy” in which water-scarce countries import large amounts of water-consuming commodities from water-rich countries to relieve existing water stress [4,9]. However, numerous studies have shown that this strategy is not reflected in international virtual water trade data. By analyzing the net effect of virtual water flows generated by the global trade in agricultural products, Ramirez-Vallejo and Rogers found that, contrary to the Heckscher–Ohlin model, virtual water trade flows are not related to water endowments [10]. Further analysis suggests that variables such as average income, population, and exports of goods and services help explain the observed changes in agricultural trade. Fraiture argues that “virtual water strategies” cannot be effective, that water resources cannot be reallocated for beneficial purposes (including relieving resource constraints), and that national trade strategies may be more influential than water scarcity [11]. Overall, the study of virtual water trade models requires not only consideration of resource endowments but also an assessment of resource endowments, and socioeconomic and political conditions. Therefore, the aim of this paper is to examine the global dimension of virtual water trade and to assess and quantify the driving forces behind virtual water exchange through global trade. More specifically, we integrate endogenous structural, socio-economic, resource endowment, and geographical distance factors to investigate how they may drive virtual water trade (VWT) over time.

From the spatial scale of related studies, VWT can be divided into global scale, regional scale, and national scale. Among them, the research on VWT at the global scale mainly includes accounting for virtual water import and export volumes in each country [12,13], analyzing global VWT link analysis [14], and the driving forces of agricultural trade pattern formation [15]. Among the existing studies, most of them focus on the regional scale and national scale. However, how to maintain the stability of the virtual water flow network structure and the balance of virtual water import and export is not only a problem that a country or region needs to face, but is also a problem to be faced by global trade collaboration and balanced regional development under the requirement of sustainable development. Therefore, it is a feasible and important research direction to analyze the international water transfer pattern from the perspective of network global evolution and then construct a VWT network and analyze its evolution mechanism.

Currently, in analyzing the drivers of VWT, scholars mostly use regression analysis, gravity models and index decomposition analysis (IDA), Structure Decomposition Analysis (SDA), and other methods to analyze the drivers of virtual water. Yang et al. used multivariate regression analysis to analyze the effects of water scarcity and other factors on water-supply-related food trade patterns in MENA countries. They found that GDP per capita, and water availability were the dominant factors explaining the variation in water-intensive crop imports during the period in question [16]. The gravity model constructed by Fracasso demonstrated that bilateral virtual water transfers are influenced by typical trade factors and environmental factors such as water endowment and water stress in the country [17]. Duarte et al. applied SDA in order to determine the role of trade in the final net water balance in Spain, a semi-arid country with periodic severe water scarcity [18]. Using a multi-regional input–output model and a log-average divisor index, Fu et al. revealed that economic and virtual water intensity effects are the most important drivers of virtual water flow changes in the EU [19]. However, there are a number of problems with existing correlation analysis methods applied to VWT drivers. First, the use of SDA and traditional index methods to analyze VWT drivers has been shown to have decomposition results that deviate from the true value. They cannot be eliminated, and the classical independence assumption limits other traditional regressions. In the VWT relationship network, countries are interdependent; that is, the existing trade links will affect the creation of a new link, so the network links are not independent, and the existing correlation analysis methods cannot deal with this situation. In addition, existing methods take less account of the time dependence of the network, i.e., the influence of the previous network topology on the current network topology. They cannot accurately reveal the network evolution path.

In this paper, we construct a VWT network consisting of 62 countries/regions and measure the contribution of different drivers to the topology of the virtual water network by constructing a time-series evolution model based on structural and temporal dependencies. To compensate for the shortcomings of traditional correlation analysis, we use Stochastic Actor-Oriented Models (SAOM) for the first time to estimate the results of “relational” data and “attribute-based” data that affect the evolution of VWT networks. We obtained comprehensive and realistic results by using SAOM to simultaneously estimate the results of “relational” and “attribute-based” data affecting the evolution of VWT networks. The paper presents an empirical analysis of the reallocation of water resources under global trade, and the related discussion of unequal trade patterns and water endowments can lead to new thinking about the rationality of global trade and the optimization of global trade patterns. The rest of the study is organized as follows: Section 2 introduces the relevant theories and hypotheses; Section 3 presents the methodology and data sources; Section 5 presents the main results; relevant conclusions and explorations will be presented in Section 6.

## 2. Theoretical Framework and Assumptions

### 2.1. Virtual Water Strategy

The “virtual water strategy” refers to water-scarce countries importing large quantities of water-consuming commodities from water-rich countries to relieve the pressure on existing water resources [4,16]. Some scholars have supported this proposal by estimating virtual water flows between countries, arguing that virtual water imports from water-rich countries in global free trade can achieve sustainable water use [20,21]. However, some studies have shown that virtual water is not reflected in international VWT data. Ramirez-Vallejo and Rogers also tested the empirical information within the conceptual framework of the Heckscher–Ohlin model and found that observed trade patterns are independent of water endowments [10]. De Fraiture argues that “virtual water strategies” cannot be effective, that water resources cannot be reallocated for beneficial purposes (including relieving resource constraints), and that national trade strategies may be more influential than water scarcity [11]. Lopez-Gunn and Llamas et al. also note that the international food trade is primarily driven by factors other than water [22]. Kumar and Singh conclude that trading strategies based solely on the virtual water concept would not help alleviate water scarcity or improve global water use efficiency [23].

In summary, this paper proposes the following Hypotheses 1.

**Hypothesis** **1a.**
*The global VWT model is not influenced by the degree of water scarcity.*


**Hypothesis** **1b.**
*The global VWT model is not affected by the degree of water abundance.*


### 2.2. Global Trade Network Drivers

The consumption of resources in one country may be used to meet the final demand of other countries, and the natural resources included in trade will certainly be transferred and redistributed among countries. Thus, each country’s resource consumption in terms of products may be different from that driven by its final demand [24,25,26]. Due to the different levels of resource scarcity between countries, the trade will increase the demand for global resources and change the global pattern of resource scarcity. Resource-scarce countries can minimize domestic environmental pressures by importing resource-intensive products from abroad. However, on the other hand, there may be cases where resource-rich countries import implied resources from resource-scarce countries, thus increasing the uneven distribution of global resources [27]. There are huge differences in the level of economic development, industrial structure, production technology, etc., between countries, and therefore there are huge differences in the efficiency of energy or resource use in each country, and all these factors will, to some extent, shape the global VWT patterns [12,28,29].

In summary, this paper proposes Hypothesis 2 about trade networks.

**Hypothesis** **2.**
*Trade networks play an important role in the formation of VWT network links.*


### 2.3. Ecologically Unequal Exchange

The concept of the world economy as a “zero-sum game” has been deeply rooted, and the economic expansion of developed countries has been at the expense of underdeveloped countries, which directly or indirectly puts enormous environmental pressure on underdeveloped countries in the process [30]. This effect will be exacerbated by the transfer of industries from developed to less developed countries in global trade [31]. The key factor driving industrial transfer is economic efficiency. The profit-driven trade expansion of developed countries offering financial investments or high-value-added goods in exchange for low-value-added goods and natural resources produced or extracted in those countries ignores the objective fact that productive activities to meet the final demand of each country require the exploitation of limited resources [32]. The first explanation for this situation comes from the “ecologically unequal exchange theory.” Trade behavior between countries with different levels of development leads to unequal exchange, and large differences in productivity lead to unequal economic configurations, which produce results favorable to developed countries and unfavorable to underdeveloped countries [33,34].

In summary, unequal trade relations negatively impact less developed countries, and, based on the ecologically unequal exchange theory, this paper proposes two hypotheses.

**Hypothesis** **3.**
*Virtual net water exporters tend to increase their export links with rich countries.*


### 2.4. Network Endogenous Structure Effect

The dynamics of trade networks are extremely complex, and the endogenous structure of the network is indispensable for predicting the formation of trade relationships. A large number of scholars have focused on the integration of exogenous mechanisms and endogenous structures of trade networks. Generally, reciprocal behavior is an important tendency in social relations, where countries in trade networks trade with other countries based on the theory of comparative advantage and reciprocity [35,36]. In addition, triangular structure (transitivity) is also considered a fundamental building block in international trade networks. The triangular structure in trade networks refers to the “friend of a friend is a friend” structure, i.e., the transmission of trade relations between partners [37]. Triadic transmission closure is an important endogenous mechanism that influences relationship choice and drives network cluster formation; forming trade links with three or more countries gives them a strong competitive advantage [38]. In addition, preference attachment is one of the important endogenous probabilistic mechanisms that drives network evolution, meaning that nodes with higher link counts in the network acquire new links faster than nodes with lower link counts, resulting in “the rich getting richer” and eventually making the nodes with higher link counts grow into hub nodes over time [39]. This mechanism has been proven in international-trade-related fields, such as trade agreement networks [40], trade dependency networks [41], and international food networks [42]. VWT networks are one of the important applications of environmental expansion input–output studies of international trade networks; therefore, the validation and discussion of the network endogenous mechanism are also applicable to VWT networks.

In summary, this paper proposes the following hypotheses on the endogenous structures that play a significant role in the evolution of trade networks.

**Hypothesis** **4a.**
*There is a reciprocal effect in the global VWT networks and plays a role in the dynamic evolution of the network.*


**Hypothesis** **4b.**
*There is a triangular structure in the global VWT networks, and there is an effect on the formation of new trade relationships.*


**Hypothesis** **4c.**
*There is a preference attachment mechanism in the global VWT networks, and there is an impact on the formation of new trade relationships.*


### 2.5. Trade Gravity Model

The gravity model is a well-known methodological framework for studying the causes of international trade. Tinbergen was the first to apply the gravity model to the study of international trade, and, in the model they developed, bilateral trade flows are proportional to GDP, which represents the size of the market in both countries, and inversely proportional to the distance between the two countries, which represents transportation costs [43]. Many empirical applications have attempted to quantify the drivers of international trade using this approach. Studies on GM and regionalization have shown that economic size and geographical distance are important drivers of trade bond formation [44,45]. It has now been increasingly used to analyze the environmental impact of international trade [46,47]. Some scholars used the gravity model global VWT drivers, and the results confirm that geographical distance has an impact on VWT [48,49].

In summary, this paper proposes the following hypotheses.

**Hypothesis** **5a.**
*Countries are more likely to establish VWT relationships with larger economies.*


**Hypothesis** **5b.**
*The more distant the geographical distance, the less likely the two sides will establish a trade relationship.*


## 3. Materials and Methods

### 3.1. Data Sources

The Environmental Extended Input–Output Table is an improvement of the Multi-Regional Inter-Country Input–Output (MRIO) that considers environmental impacts. It is mainly used to describe the direct water consumption by the production process, and the indirect water consumption triggered to meet the final demand in other areas by constructing water satellite use accounts to be integrated with the existing MRIO database. The most common extension is the addition of energy consumption and energy use or energy flows to the traditional MRIO. In order to get the latest global VWT volumes as accurately as possible, two types of data are combined and transformed: value-physical input–output tables and water consumption by sector. The data sources are as follows.

Multi-Regional Inter-Country Input–Output (MRIO) data provide relevant quantitative information at sufficient granularity to describe and analyze production and trade activities. To meet certain information and analysis needs related to Asia and the Pacific, the Asian Development Bank (ADB) added more details on Asian economies to the World Input–Output Database (WIOD), resulting in the ADB Multi-Regional Input–Output Database (ADB-MRIO) (https://data.adb.org/; accessed on 18 February 2022). In the ADB-MRIO, 62 countries and regions and the Rest of the World (RoW) are included between 35 industrial sectors in input–output relationships (Appendix A). In this paper, trade data are taken for three time periods, 2000, 2010, and 2015, and the 35 sectoral trade values are combined and aggregated to the national level. Virtual water use data are collected from the EORA database (https://worldmrio.com/; accessed on 18 February 2022), which includes 35 environmental indicators, including air pollution, energy use, greenhouse gas emissions, water use, and land occupation. In the EORA database, all environmental data are systematically assigned to different sectors in different countries. The database contains green, blue, and gray water usage for 190 countries (regions) from 1990 to 2015. In this paper, the water use data of 62 countries/regions are selected, and the data of green, blue, and gray water are integrated to obtain the country’s water use in different years. This water use data are used to convert water resources expansion input–output data.

In this paper, referring to the research results of related scholars, the global VWT network is used as the explanatory variable, and eight indicators are selected as explanatory variables from four dimensions: economic, social, water resources, and geography (see Table 1). The data of the indicators were selected in 2014 as the time cross-section, and the data sources include World Development Indicators (WDI) (https://databank.worldbank.org/home; accessed on 21 March 2022) and the Center for International Prospective Studies (CEPII) (http://cepii.fr/; accessed on 21 March 2022).

### 3.2. VWT Imbalance Index

In recent years, some scholars have used MRIO to measure the transfer of hidden water resources in trade. The regional input–output model combined with water resources input can quantify the consumption of water resources within a region (virtual water footprint) and the transfer of water consumption between regions (VWT). This paper draws on the construction method of the Inter-Regional Input–Output Model of Water Resources of Lenzen et al. and, based on the MRIO table, an inter-country water resource input–output table [50].

First we calculate the water use coefficient:(1)$$w^{r}=\frac{W^{r}}{X^{r}}$$
where the superscript r represents a country (region), W represents the total water consumption from EORA environmental data, and X represents the total output of ADB-MRIO. The MRIO balance formula is written as:(2)x=Ax+y
where A is m×m MRIO technical coefficient matrix (*m* is the number of countries/region), Y represents the final demand matrix of m×m, and X denotes the total output column vector of m×1. As shown below:$$A=\left[\begin{array}{cccc} A^{11} & A^{12} & \cdots & A^{1n}\\ A^{21} & A^{22} & \cdots & A^{2n}\\ \vdots & \vdots & \ddots & \vdots \\ A^{n1} & A^{n2} & \cdots & A^{nn} \end{array}\right];Y=\left[\begin{array}{cccc} y^{11} & y^{12} & \cdots & y^{1n}\\ y^{21} & y^{22} & \cdots & y^{2n}\\ \vdots & \vdots & \ddots & \vdots \\ y^{n1} & y^{n2} & \cdots & y^{nn} \end{array}\right];x=\left[\begin{array}{c} x^{1}\\ x^{2}\\ \vdots \\ x^{n} \end{array}\right];$$

The above formula can be rewritten as
(3)$$x={\left(I-A\right)}^{-1}y=Ly$$
where ${\left(I-A\right)}^{-1}$ is the Leontief inverse matrix. In this paper, referring to the research of Jiang and his colleagues [51] as well as Duarte and his colleagues [48], the water coefficient is directly multiplied by the diagonal matrix W, the Leontief inverse matrix L and the final matrix Z to obtain the virtual water transaction matrix H.
(4)H=WLZ

Through this formula, the virtual water input–output matrix data can be obtained. The diagonal elements of the matrix are the virtual water consumption of products produced by the country (region); the non-diagonal element is the virtual water import and export trade. In this paper, the diagonal elements are reduced to 0 to calculate each country’s virtual water import and export volume. This paper uses each country’s virtual water export trade volume/virtual water import trade volume to obtain WTI, which is used to express the country’s VWT imbalance and takes the natural logarithm of WTI, so $\ln\left(WTI\right)=\ln\left(\frac{Water\;exports}{Watre\;imports}\right)$. A value of $\ln\left(WTI\right)$ greater than 0 indicates that the country is a virtual water exporting country while less than 0 indicates that the country is an importing country. These data are then converted into a ranking value from 1 to 10 to adapt to the data limitations of the stochastic modeling framework.

### 3.3. Construction of Dynamic Evolution Model of VWT Network

The Stochastic Actor-Oriented Model (SAOM) is used to study the dynamic evolution of social networks between two or more discrete points in time and was developed by Tom Snyders and colleagues [52]. The SAOM model provides a stochastic modeling environment for analyzing longitudinal network data that allows simultaneous analyses of network dynamic evolution and changes in network node behavior and is therefore recognized as one of the current research network dynamic evolution [53]. The SAOM model treats network evolution as the process of establishing, continuing, or terminating connections of network nodes, and a rate function determines the changes in trade relationships in VWT networks. The SAOM model uses a logistic regression model to model the probability of selection. The expression of the rate function is rewritten as:(5)$$p_{i}\left(x_{0},x,v,w\right)=\frac{\exp\left[f_{i}\left(x_{0},x,v,w\right)\right]}{\sum_{x^{\prime }\in C\left(x_{0}\right)}\exp\left[f_{i}\left(x_{0},x^{\prime },v,w\right)\right]}$$
where $x_{0}$ represents the initial state of the network, x represents the potential new state of the network, v represents the individual attributes in the network, and w represents other extrinsic attributes, including distance, etc.

The SAOM model defines a utility function as the objective function of the actor, which can allocate trading partners for the actors (countries/region) in the VWT networks according to the maximum principle, and simultaneously analyzes the impact of the endogenous structure of the network as well as the extrinsic subject characteristics on the trade network connection. The utility function expression is rewritten as:(6)$$f_{i}\left(x_{0},x,v,w\right)=\underset{k}{\sum }\beta_{k}e_{ki}\left(x_{0},x,v,w\right)$$
where $\beta_{k}$ is the estimated parameter and $e_{ki}$ is the various factors affecting the dynamic evolution of the network.

In SAOMs, the evolutionary network process is portrayed by two functions, the utility equation and the rate equation. In the objective evolutionary equation, the explanatory variables are classified into five types (Table 2). One is the network basic effect, which includes two variables: density effect and reciprocity. The second is the degree centrality effect, including the out-degree activity effect, the in-degree popularity effect, and the out-degree popularity effect, three variables, to test the influence of the preference attachment mechanism on the evolutionary characteristics of VWT networks. Third, the tripartite relationship effect is used to test the possible impact of the network closure mechanism on the VWT network. The fourth is the geographical location effect, to test the effect of geographic distance on network relationships in a fluid spatial environment. Fifth, actor-relationship effects are used to examine the effect of country attribute characteristics on the propensity of countries to issue.

The SAOM model allows examination of which factors significantly influence the formation of trade relationships during the dynamic evolution of VWT and overcomes the shortcomings of traditional econometric models, enabling the estimation of changes in network data. This paper uses the R version of the Simulation Studies in Empirical Network Analysis (SIENA) software program to implement the modeling and follows the model fitting and convergence procedures specified by Ripley et al. [54].

## 4. Results

### 4.1. Global Virtual Water Flow Analysis

In order to further analyze the virtual water flow problem, this paper visualizes the virtual water trading network. As shown in Figure 1, the importance of a node is measured by the number of shortest paths through that node; the more significant the node, the greater the state’s role in the trade network. The larger nodes are China, the United States, and Germany, which present a tripartite situation. From 2000 to 2015, China’s position in the VWT network continued to rise, becoming a key link in the trade network. In general, the central nodes of the VWT network are divided into two categories: emerging economies such as China, India, and Russia, and industrialized countries such as the United States, Japan, and Germany. From the perspective of virtual water flow, there is often a large trade flow between developed and developing countries. The largest VWT flow occurs between China and the United States, and the VWT flow is as high as 47352.48Mm3. The import and export volume of virtual water in middle-income countries is much higher than in other countries. The VWT data (Appendix B) show that, in 2015, middle- and upper-income countries exported 369.96Gm3 to high-income countries, more significant than the 194.844Gm3 exported by lower-income countries. Therefore, it can be seen that a large amount of virtual water flows from these emerging economies to developed countries.

As shown in Figure 2, the top countries with the largest virtual water import trade among the net virtual water importing countries are the United States, Germany, Japan, and the United Kingdom, all of which are developed countries. Similarly, the top countries with the largest virtual water export trade among the net virtual water exporting countries are China, India, Russia, and Mexico, all of which are emerging developing countries. In addition, the largest virtual water flows from developing to developed countries. Net importing countries, in addition to high-income countries, include some less developed countries, including Brunei, Bhutan, and Nepal; these countries have a poor industrial base. Many products rely on imports and therefore show high virtual water imports. Canada is a net exporter of virtual water due to the fact that Canada exports a large amount of resource-based products, which have a large water-use factor, thus resulting in the export of virtual water. Looking at the graph of available freshwater per capita for 63 countries, China does not have a high amount of fresh water per capita, but it is the largest virtual water exporter. Countries in the same situation include India, Mexico, and Vietnam, and, most notably, Indonesia, which does not have the advantage of water resources and has a large VWT import/export deficit. In contrast, countries such as the United States, Australia, and Norway, which have more abundant water resources, still become net virtual water importers. In addition, Russia, Canada, etc., as water-rich countries, maintain a large number of virtual water exports.

### 4.2. Analysis on the Driving Factors of VWT

Four independent models are given in Table 3, where M1 is the base model, including some endogenous structural effects and control variables such as geographical distance, logistics facilities, population size, and country income classification. Different models focus on testing different hypotheses, with M2 testing hypothesis 1, M3 testing hypotheses 2 and 3, and M4 focusing on hypothesis 4. The four models are set up to provide robustness checks for the results. The GOF test of the VWT-driven evolution model proposed in this paper is shown in Appendix C. The fitting results of all models are far greater than 0.05, which proves the extremely high applicability of the model.

First, the VWT network’s evolution rate parameters in M1–4 all show smaller changes in the second period (2010–2015) than in the first period (2000–2010), which implies that there are more opportunities to form trade links in the first period. Overall, the global trade network is still in a period of adjustment, and the network structure is not yet stable.

Model 1 and model 4 analyze the endogenous network structure’s driving role in forming VWT networks. The model results support hypothesis 4c that the preference dependency effect promotes the evolution of VWT. The results reveal the mechanism of network endogenous structure theory on the evolving structure of the VWT network over a 15-year period. Out-degree activity (sqrt) and in-degree popularity (sqrt) fit parameters in SAOMs are positive and significant. This analysis provides empirical evidence for the existence of preference attachment mechanisms in the evolution of national/regional degree centrality systems in VWT networks. The positive parameter of out-degree activity (sqrt) represents the preference of this virtual water importing country to maintain export relationships with other countries, while the positive parameter of in-degree popularity (sqrt) represents the preference of virtual water importing countries to maintain import relationships with other countries. The positive parameter of out-degree activity (sqrt) represents the preference of this virtual water exporting country to maintain export relationships with other countries. In contrast, the positive parameter of in-degree popularity (sqrt) represents the preference of virtual water importing countries to maintain import relationships with other countries. Out-degree popularity (sqrt) has a negative and significant parameter, indicating that countries sending relationships on a larger geographical scale do not imply receiving relationships on a larger geographical scale. From the perspective of VWT networks expansion, it refers to the fact that virtual water exporting countries do not tend to maintain a large number of virtual water importing links.

The model results support hypotheses 4a and 4b that reciprocity and triangular-type structures contribute positively to the VWT network. The coefficients of reciprocity and transitive ties in SAOMs are positive and significant, indicating that VWT networks have more reciprocal and transitive linkages than random networks. The coefficient of transitive triplets is positive, while the fitted coefficient of 3-cycles is negative, indicating that network closures are in the form of transitive closure rather than cyclic closure. The analysis results are consistent with the previous results of the fitting of the out-degree convergence effect, which indicates that there is not only a global trend of hierarchical differentiation in the VWT networks but also a local trend of hierarchical differentiation. Reciprocal links and network-closure mechanisms can be understood as observable results of national trade networks sharing scarce resources and reducing transaction costs. Since key resources are concentrated in a few countries, more linking relationships will occur between these countries, thus creating an increasing number of triangular relationships. In summary, hypothesis 4 holds, and all the endogenous structural effects of the network mentioned in this paper constitute the driving mechanism for the growth and development of the virtual water network and the basis for the complexity and orderliness of the network.

There is a meritocratic linking mechanism in the growth and development of the global VWT network pattern. The fitted coefficients of GDPtot alter in models 3 and 4, income alters, and Poptot alter variables in models 1 and 2 are positive and significant, indicating that cities with larger economies and higher per capita incomes tend to receive relationships on a larger geographical scale when link strength and breadth are considered together. GDPtot similarity has a negative and significant fit coefficient, indicating that, from the perspective of link breadth alone, there are more linking relationships between countries with larger economic size disparities compared with the desired value.

The model results show that hypothesis 5a is supported, and hypothesis 5b is overturned. The negative and significant fit coefficients of the distance variable in Models 1–4 support the idea that distance has an impact on the establishment of trade relationships and is consistent with the traditional trade gravity model that states that the closer the distance, the greater the likelihood of trade ties between countries. The interaction termint. GDPtot alter x distance in models 3 and 4 with negative parameters, and significance does not match the traditional trade network gravity model findings. After controlling for other network variables, the fact that countries have trade links with smaller economies is evidence of the current expansion of trade globalization to broader markets. In summary, hypothesis 5 proposed in this paper regarding the gravity model in VWT is partially valid. The traditional gravity model theory of trade has reduced applicability in today’s accelerated expansion of global trade networks.

The results of models 3 and 4 show that hypothesis 3, which is based on the ecologically unequal exchange, is supported. Specifically, the fitted coefficient of the explanatory variable WTI ego is positive and significant, representing that higher WTI values tend to send links in VWT. Therefore, hypothesis 3 is supported and virtual water net exporting countries will maintain virtual water export trade relationships. The fitted coefficient of WTI ego x GDPtot alter is also positive and significant, further supporting the ecologically unequal exchange theory. Countries establishing VWT relationships with rich countries tend to become net virtual water exporters, with large virtual water flows to developed countries. The results show that the formation of VWT patterns and the level of national economies have some correlation.

Models 3 and 4 were validated for the influence of the trade network on the global VWT network, and the results show that hypothesis 2 is supported. Specifically, the trade network fitting parameter is positive and significant. Economic trade brings about changes in virtual water; water resources flow from the place of output to the place of use with the trade of products, forming a virtual water reallocation. The change in global trade is mainly reflected in the growth of trade volume and trade structure. In the case of a constant trade structure, the increase in total trade volume will undoubtedly lead to more global virtual water consumption. If the trade structure changes so that more products are produced in regions with less water scarcity while the final demand and demand structure of countries around the world remains unchanged, the structural change may lead to a decrease in overall global water consumption. Thus, the trade network itself must have a significant impact on virtual water flows.

The coefficient of Water stress ego fit in model 2 is not significant, while the coefficient of Waterpc ego fit is negative and significant, so hypothesis 1a holds, while hypothesis 1b is overturned. The results indicate that global VWT is not influenced by the degree of national virtual water scarcity but, to some extent, by the amount of water resources per capita. However, countries with sufficient water resources per capita are only e^0.008^ times more likely than other countries to send links, i.e., virtual water exports, less likely than other factors to drive VWT networks.

## 5. Discussion

From the results of the global VWT networks driving force model, the global trade pattern and ecologically unequal exchange theory proposed in this study were validated, and the water endowment hypothesis, and the traditional trade gravity model were partially supported.

Firstly, by setting the global trade network as the explanatory variable, the analysis using the latest network econometric model concluded that trade patterns have a great driving effect on the virtual water flow situation, and this conclusion was also confirmed in previous studies [55,56,57]. In the past, many scholars tried to simulate the impact of global trade on resources and the environment by setting up a “zero trade scenario.” However, since trade not only changes the original production location of products but also stimulates global consumption and changes the input–output structure among national sectors, the real “zero trade scenario” is difficult to simulate and is not realistic and feasible. In contrast, this paper takes trade networks as explanatory variables, covering both trade flows as well as trade structures, and confirms the driving force of trade on virtual water flows.

Secondly, hypothesis 3, proposed in the paper based on unequal exchange of production, is supported. Therefore, we argue that there are some established patterns of virtual water trading under global trade. Less developed countries that form trade relationships with developed countries tend to have higher WTI. Countries with high WTI export large amounts of virtual water intensive products and further expand their virtual water exports in order to maintain comparative advantages in production activities that meet the foreign demand. Countries with high WTI export a large number of virtual water intensive products and further expand their virtual water exports in order to maintain a comparative advantage in production activities to meet foreign demand, and the above assumptions are confirmed in the model. Copeland and Taylor’s study points out that developed countries exchange financial investments or high-value-added goods for low-value-added goods and natural resources produced or extracted in these countries [58]. Several studies have confirmed the tendency of developed countries to transfer environmental pressures or pollution implicit in the production process to developing countries through international trade. The imbalance between economic gains and virtual water use has also been found in VWT studies, which show that western developed economies such as the United States, Japan, and Germany are net virtual water importers; in contrast, developing countries such as China and India are net virtual water exporters [59,60]. Emerging developing countries are seeking to climb up the global value chain and shift the industrial chain to high-value-added capital-intensive industries. However, to maintain their global market share, less developed countries have to assume the final demand for natural resource intensive products from developed countries, thus increasing the environmental pressure on their own countries. This is an important feature of the carefully orchestrated “decoupling” of resource consumption and economic growth in less developed countries by developed countries. In general, the existing trade patterns of the global VWT have negative effects. Therefore, for deep participation in global trade, countries should take into account their own economic situation and the future needs of industrial transformation and development, and coordinate and complement domestic production with the foreign industrial layout to ensure rapid economic development while reducing domestic resource consumption.

In addition, hypothesis 4 is accepted in that the endogenous structure of the network has a positive driving effect on the VWT, which is a completely new and significant finding in this paper. In this paper, through an empirical analysis of VWT networks in 62 countries, we found that meritocratic links, path dependence, reciprocity, and transmissibility links have an extremely strong explanatory power for the evolutionary development of virtual water networks over 15 years. Christina et al. found that reciprocity and triangular-type structures influence global trade networks [61]. Cohen et al. found that by analyzing Chinese rice; they found the importance of reciprocity, transitivity, and universal exchange in the choice of friends of rice producers after analyzing the network of relationships among rice producers in China [62]. Their study pointed out that the choice of individuals in the network about whether to be associated with others drove the network’s evolution. While the important role of endogenous structure has been found in all other network analyses, this paper is the first to address the role of endogenous structure in global VWT networks, validating the strong influence of structural dependence on VWT networks.

The role played by traditional trade models in VWT is tested in the paper, and hypothesis 5 is partially supported. In the analysis of the control variables’ geographical distance, hypothesis 5b is rejected, implying that the traditional gravity model of trade is overturned and the trade network is expanding to more distant and larger markets. Developing countries that benefit from free trade further expand their global influence [63,64]. In the case of China, trade relations with developing and less developed countries with small economies are of economic and political importance. In terms of trade exports, there is a serious bilateral trade imbalance between China and less developed countries, and the share of China’s exports to them is much larger than the share of imports. From the analysis of comparative advantage theory, China will move to industries with more pronounced economies of scale in the future and may shift industries with a lower value added to less developed regions, including agricultural products and natural resources [65]. In summary, the trade objects of global VWT are not completely influenced by geographical distance, and the traditional gravity model theory of trade is less applicable in today’s accelerated expansion of global trade VWT networks.

We propose hypothesis 1a and hypothesis 1b based on the resource endowment theory, which stems from the controversy in past studies about whether water endowments have an impact on VWT. Some scholars have tried to verify whether water-rich countries are net virtual water exporters in global trade [23,66,67], and the final results show that this is not the case. Many water-rich countries are net virtual water importers, and water-scarce countries are net virtual water exporters, which is also verified in the figure of this paper. Other scholars have used econometric models to analyze whether water endowments have an impact on national VWT flows [16,17,18] and have concluded that water endowments have an impact on water trade demand. This paper disentangles water endowment into water scarcity and water abundance, and the results confirm that water scarcity has no impact on the evolution of VWT networks, while water abundance affects virtual water flows, although this driver is weaker than other important factors. This paper answers the question of whether water endowments affect VWT from a new perspective, enriching the existing empirical research on VWT. 

## 6. Conclusions

Using virtual water input–output data for 62 countries from 2000–2015, this paper finds that high-income countries tend to be net virtual water importers by analyzing virtual water imports and exports between countries, with upper-middle-income countries importing a large amount of virtual water while emerging developing countries undertake a large number of virtual water exports. This paper addresses the most popular hypotheses of VWT. It uses the latest SAOM to develop a model of the drivers of VWT that includes both univariate and binary variables and associated structural effects, confirming that countries that have established VWT relationships with rich countries tend to be net virtual water exporters and that these countries will still tend to export virtual water in the future, indicating an ecologically unequal exchange. In addition, this paper finds that water abundance impacts VWT, and that trade networks drive this more than the amount of freshwater per capita. The trade gravity model is partially supported by the finding that trade networks are expanding to more distant and larger markets. Most importantly, this paper points out for the first time the influence of the endogenous structure of the network on VWT relationships, and the results confirm that meritocratic links, path dependence, reciprocity, and transmissive links have strong explanatory power for the evolutionary development of virtual water networks.

The development of global resource trade has an important impact on the economies and resource environment of all countries. A reasonable trade structure and change trend are conducive to the coordinated development of global economies and promote the balanced distribution of resources. Therefore, it is important to grasp the current global trade implied resource and environment transfer for the coordinated and sustainable development of the global economy and resource environment. We not only conducted a study on global trade implied water transfer based on the current trade situation but also focused on several important related factors, such as the endogenous structure of trade, resource endowment, and economic dimension, and delved into the driving factors behind the global virtual water trade. By exploring the current trends in trade structure that are not conducive to resource conservation and balanced global resource distribution, we provide a realistic basis for relevant national trade policies.

The analytical framework provided in this paper can break through the independence assumptions while estimating results for multiple environmental economics theories, enabling an in-depth analysis of the evolution of network structures. This paper examines water flows from the perspective of global trade, where the water impacts associated with the expanding international trade have been increasing. Not only has the ecologically unequal exchange between developed and less developed countries given rise to many discussions, but also the rapid population and economic growth in developing economies and the water-intensive consumption patterns in industrialized regions constitute a concern for the current and future sustainability of the globalization process. Therefore, this paper emphasizes that a multidisciplinary analytical framework in water resources management, through the integration of resource trade with long-term economic development studies, is essential to lay the groundwork for water conservation worldwide. Although the approach of bringing the trade network as a whole into the model as an independent variable maximally preserves the flow and network structure of global trade, it is not possible to explore in the paper the mechanism by which the global trade network structure exacerbates national virtual water imbalances, which is a limitation of this paper. Therefore, we will conduct further research from that perspective later.

## Figures and Tables

**Figure 1 ijerph-20-03234-f001:**
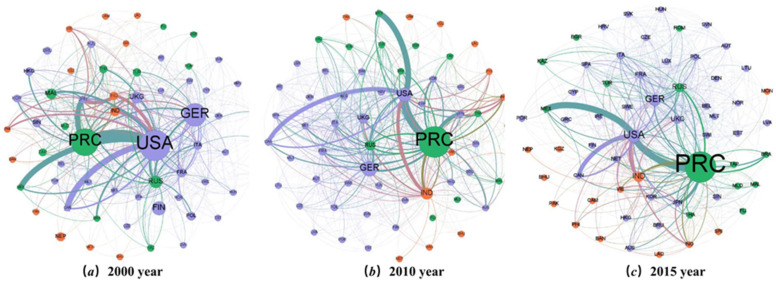
VWT network. Note: The color of the nodes represents the income level. Blue represents high-income countries; green represents middle-income countries; and red represents lower-income countries. The size of the node represents the centrality of the country in the trade network.

**Figure 2 ijerph-20-03234-f002:**
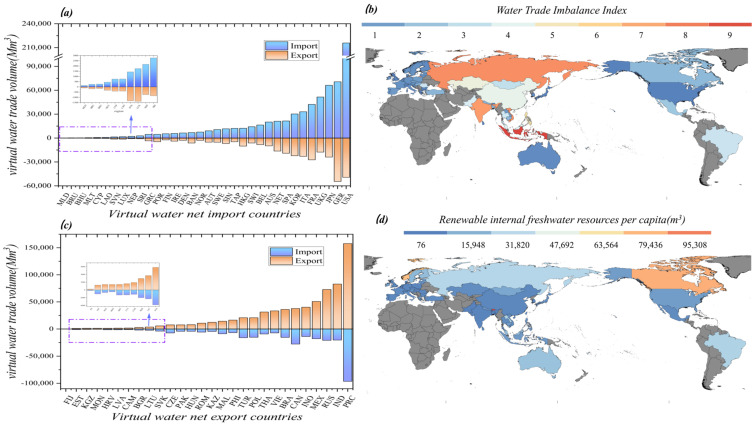
(**a**) Virtual water import and export volume of virtual water net importing countries/region; (**b**) virtual water import and export volume of virtual water net exporting countries; (**c**) WTI value of 62 countries/region (WTI value less than 1 is virtual water net importing country, if it is greater than 1, it is a virtual net exporter of water); (**d**) The per capita available fresh water in 62 countries/regions.

**Table 1 ijerph-20-03234-t001:** Variable selection and variable symbols of the driving force model.

Dimension	Variable Indicator	Variable Explanation	Quantity Unit	Variable Symbol
Economic Dimension	Gross domestic product	It reflects the level of national economic development	U.S. Dollar	GDPtot
	Country income classification	It reflects the level of national economic development	U.S. Dollar	income
Social Factors	Logistics performance index	Trade and Logistics Infrastructure Score (Low 1~High 5)	-	LPI
	Input–output trade network	It reflects the import and export of the global trade network	-	Trade
	Population size	The larger the population, the higher the demand for goods	people	Poptot
Water Endowment	Water crunch rate	It reflects the degree of shortage of national water resources	%	waterstress
	Freshwater resources per capita	It reflects the wealth of the country’s water resources	m^3^	waterpc
Geographical Factors	Distance between countries	It reflects the relative geographical situation between countries	km	Distance

**Table 2 ijerph-20-03234-t002:** Network effects in SAOMs.

Endogenous Network Effects [SIENA Code]
Outdegree effect [density]	∑jxij	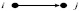	Whether global trade network relationships are random processes.
Reciprocity [recip]	∑jxijxji	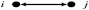	Whether there is a reciprocal connection between countries i and j.
Transitive ties [transTies]	∑hxihmaxjxijxji	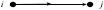	Whether there are transit links in the trade network.
Transitive triplets [transTrip]	∑j,hxihxijxjh	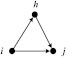	Whether there are transitive closed links in the trade network.
**3-cycles[cycle3]**	∑j,hxijxjhxhi	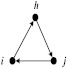	Whether the network link relationship has cyclic closed links.
Outdegree activity (sqrt) [outActSqrt]	xi+xi+	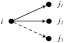	Cities with higher out-degree values tend to send more ties.
Indegree popularity (sqrt) [inPopSqrt]	∑jxijx+j	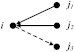	Countries with higher in-degree values tend to receive more ties.
Outdegree popularity (sqrt) [outPopSqrt]	∑jxijxj+	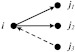	Countries with higher out-degree values tend to receive more ties.
Network formation effects [SIENA code]
covariate-ego [egoX]	vnixi+		Countries with higher attribute value X tend to send relationships
covariate-alter [altX]	∑jxijvnj	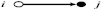	Countries with higher attribute value X tend to receive ties
covariate similarity [simX]	∑jxijsimijvn−simvn	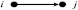	Countries with similar attribute value X tend to build relationships
Dyadic geographic distance [distance]	∑jxijwij−w¯	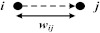	Whether geographic distance limits the sending relationship.

**Table 3 ijerph-20-03234-t003:** Estimated results of global VWT network drivers.

	Evolution of Trade Networks	Model 1		Model 2		Model 3		Model 4	
	VW TRADE rate (period 1)	5.036 *	(0.407)	4.785 *	(0.353)	5.984 *	(0.540)	6.213 *	(0.550)
	VW TRADE rate (period 2)	3.059 *	(0.270)	2.995 *	(0.261)	3.340 *	(0.309)	3.399 *	(0.316)
	outdegree (density)	−4.236 ***	(0.608)	−1.485 ***	(0.102)	−4.134 ***	(0.630)	−4.953 ***	(0.307)
**H4a**	reciprocity	1.405 ***	(0.137)	1.602 ***	(0.132)	1.294 ***	(0.143)	1.410 ***	(0.164)
**H4b**	transitive triplets	0.145 ***	(0.016)			0.139 ***	(0.018)		
	transitive ties	2.127 ***	(0.603)			1.891 ***	(0.625)		
	3-cycles	−0.079 ***	(0.020)			−0.059 ***	(0.021)		
**H4c**	indegree—popularity (sqrt)							0.664 ***	(0.092)
	outdegree—popularity (sqrt)							−0.197 *	(0.098)
	outdegree—activity (sqrt)							0.365 ***	(0.041)
	Distance	−0.008 ***	(0.002)	−0.010 ***	(0.002)	−0.008 ***	(0.002)	−0.015 ***	(0.002)
**H2**	Trade network					0.982 ***	(0.142)	1.031 ***	(0.150)
	LPI ego	−0.012 *	(0.004)	−0.324 **	(0.116)				
	LPI alter	0.069	(0.094)	0.258 **	(0.087)				
	Poptot ego	0.012 ***	(0.004)	0.018 ***	(0.005)				
	Poptot alter	0.015 ***	(0.004)	0.023 ***	(0.003)				
	income alter	0.242 *	(0.113)	0.549 *	(0.095)				
	income ego	0.195 †	(0.111)	0.187 †	(0.114)				
	income similarity	0.040 *	(0.136)	0.010	(0.127)				
	WTI alter					−0.119 ***	(0.031)	0.080	(0.052)
	WTI ego					0.121 ***	(0.039)	0.128 ***	(0.032)
	WTI similarity					0.151	(0.146)	0.133	(0.145)
**H5a**	GDPtot alter					0.237 ***	(0.048)	0.190 ***	(0.054)
	GDPtot ego					0.087 *	(0.085)	0.001	(0.049)
	GDPtot similarity					−0.500 ***	(0.143)	−0.232 †	(0.135)
**H1a**	Water stress ego			0.012	(0.008)				
**H1b**	Waterpc ego			−0.008 †	(0.004)				
**H3**	int. WTI ego x GDPtot alter					0.003 *	(0.001)	0.003 **	(0.001)
**H5b**	int. GDPtot alter x distance					−0.075 ***	(0.021)	−0.041 *	(0.019)

Note: †, *, ** and *** respectively represent the significance levels of 10%, 5%, 1% and 0.1% in the two-tailed test. Standard errors are in parentheses.

## Data Availability

Not applicable.

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
