# Peer review of "Structural Characteristics and Evolutionary Drivers of Global Virtual Water Trade Networks: A Stochastic Actor-Oriented Model for 2000–2015"

_ijerph, 2023, doi:10.3390/ijerph20043234_

Round 1
Reviewer 1 Report
The authors aim to estimate the main structural characteristics and evolutionary drivers related to the global virtual water trade networks over a 15-years period. The work is original and it overcomes a gap in the field, specifically by a methodological perspective.
Findings suggest that virtual water flows from less developed countries to developed countries under some trade conditions. This confirms some other study on this item. Furthermore, the authors found that unequal trade affects consumption of virtual water in less developed countries, where it is excessively consumed.
I suggest to improve the part on policy implications for better understanding the relevance of the study on a planetary scale.
No additional comments on the tables and figures that appear appropriated and information is correctly reported.
Over a 15-years period, the study is focused on estimating the structural characteristics and the drivers of global virtual water trade networks. The study is really original and the subject - Virtual Water Trade - and the proposed method - Stochastic Actor-Oriented Model - are fully suitable with respect the objectives.
The paper is surely publicable in the journal. The findings are very interesting and their implications should be commented more in depth.
Author Response
Response to Reviewer 1 Comments
Thank you for your letter dated Jan 4, 2023. We thank the reviewers for the time and effort they have put into reviewing the previous version of the manuscript. Their suggestions have enabled us to improve our work. Based on the instructions provided in your letter, we have made a revised version of the manuscript.
Appended to this letter is our point-by-point response to the comments raised by the reviewers. The comments are reproduced, and our responses are given directly afterward in red color.
We would like also to thank you for allowing us to resubmit a revised copy of the manuscript. And we hope that the revised manuscript could be accepted for publication in International Journal of Environmental Research and Public Health.
Thank you for your important and kind review comments.
Sincerely
Lizhi Xing
Point 1: I suggest to improve the part on policy implications for better understanding the relevance of the study on a planetary scale.
Response 1: Thank you very much for your suggestions regarding the policy implications. We have focused on the policy implications of the study findings in the global context in the conclusion section, as shown in the revised section below.
The development of global resource trade has an essential impact on the economies and resource environment of all countries. A reasonable trade structure and changing trends are conducive to the coordinated development of global economies and promote a balanced distribution of resources. Therefore, it is vital to grasp the current situation of global trade implied resource and environmental transfer for the coordinated and sustainable development of the global economy and resource environment. We not only conduct a study on global trade implied water transfer based on the current trade situation but also focus on several critical related factors, such as the endogenous structure of trade, resource endowment, and economic dimension, and delve into the driving factors behind the global virtual water trade. By exploring current trends in trade structures that are not conducive to resource conservation and balanced global resource distribution, we provide a realistic basis for relevant national trade policies.
Thank you very much!

Reviewer 2 Report
1. Introduction
The scientific issues authors intend to address are not somewhat explicit in this section,so are the objectives of this manuscript. Additionally, authors stated that this manuscript designed to compensate for the shortcomings of traditional correlation analysis, however, they actually failed to fully review existing works and pointed out what the the shortcomings of traditional analysis are.
2. Theoretical framework and assumptions
What is the difference between Hypothesis 3a and 3B, and what are implications of such a difference?
3. Materials and Methods
Authors need to describe more details in data sources. I believe present information is inadequate to inform what information, where and how you obtain, especially the information on the virtual water consumption and exchange among trade partners. Additionally,authors have to provide 62 countries and regions and 35 industrial sectors as an appendix.
In table 2, there are both cities and countries in different connections, however I believe they have different roles in global trade for cities and countries, so under what the circumstances at the centre of these trade connections are for cities or countries?
4. Results
The organization for subsection 4.2. Analysis on the driving factors of VWT is not somewhat logically explicit, please reorganize this subsection in the way that explicitly accept or reject your hypotheses.
5. Discussion
Authors should more explain what accepting or denying the hypotheses implies.
Finally, there are some sentences difficult to understand, which need authors to thoroughly edit them.
Author Response
Response to Reviewer 2 Comments
Thank you for your letter dated Jan 4, 2023. We thank the reviewers for the time and effort they have put into reviewing the previous version of the manuscript. Their suggestions have enabled us to improve our work. Based on the instructions provided in your letter, we have made a revised version of the manuscript.
Appended to this letter is our point-by-point response to the comments raised by the reviewers. The comments are reproduced, and our responses are given directly afterward in red color.
We would like also to thank you for allowing us to resubmit a revised copy of the manuscript. And we hope that the revised manuscript could be accepted for publication in International Journal of Environmental Research and Public Health.
Thank you for your important and kind review comments.
Sincerely
Lizhi Xing
Jan. 31th, 2023
Email: itwasa@163.com
Point 1: Introduction
The scientific issues authors intend to address are not somewhat explicit in this section,so are the objectives of this manuscript. Additionally, authors stated that this manuscript designed to compensate for the shortcomings of traditional correlation analysis, however, they actually failed to fully review existing works and pointed out what the the shortcomings of traditional analysis are.
Response 1: Thank you for pointing out the shortcomings of this part. The main issue addressed in this paper is the desire to propose a comprehensive analysis of the structural evolution and drivers of the global virtual water trade network at the global scale, which is essential for the coordinated and sustainable development of the global economy and the resource environment. We apologize for not elaborating on this goal in more detail in the paper, and we have made changes. The scientific aspects of the study have been elaborated and revised as follows.
In order to achieve water savings in global commodity exchange, Tony Allan first proposed a "virtual water strategy" in which water-scarce countries import large amounts of water-consuming commodities from water-rich countries to relieve existing water stress [1]. However, numerous studies have shown that this strategy is not reflected in international virtual water trade data. By analyzing the net effect of virtual water flows generated by the global trade in agricultural products, Ramirez-Vallejo and Rogers found that, contrary to the Heckscher-Ohlin model, virtual water trade flows are not related to water endowments [2]. Further analysis suggests that variables such as average income, population, and exports of goods and services help explain the observed changes in agricultural trade. Fraiture argues that "virtual water strategies" cannot be effective, that water resources cannot be reallocated for beneficial purposes (including relieving resource constraints), and that national trade strategies may be more influential than water scarcity [3]. Overall, the study of virtual water trade models requires not only consideration of resource endowments but also an assessment of resource endowments and socioeconomic and political conditions. Therefore, this paper aims to examine the global dimension of virtual water trade and to assess and quantify the driving forces behind virtual water exchange through global trade. More specifically, we integrate endogenous structural, socioeconomic, resource endowment, and geographical distance factors to investigate how they may drive VWT over time.Allan J A. Water use and development in arid regions: Environment, economic development and water resource politics and policy[J]. Review of European Community & International Environmental Law, 1996, 5(2): 107-115.
- Allan J A. Fortunately there are substitutes for water otherwise our hydro-political futures would be impossible[J]. Priorities for water resources allocation and management, 1993, 13(4): 13-26.
- Ramirez-Vallejo J, Rogers P. Virtual water flows and trade liberalization[J]. Water Science and Technology, 2004, 49(7): 25-32.
- De Fraiture C, Cai X, Amarasinghe U, et al. Does international cereal trade save water?: the impact of virtual water trade on global water use[M]. Iwmi, 2004.
In addition, this paper focuses on two directions to make up for the shortcomings of existing studies. One is the scope of the study: the existing studies on virtual water flows at the global scale are small, focusing mainly on the volume of virtual water trade in individual countries. In order to fill this gap, we take 62 countries/regions around the world covering different income countries as the research object and analyze the drivers behind the virtual water trade from the global scale; Second, the research method: in the existing research related to the drivers of virtual water trade network, the use of SDA and IDA to analyze the drivers of virtual water trade has been proven to have decomposition results deviating from the actual value. The classical independence assumption limits the other traditional regressions. However, in the virtual water trade network, there are interdependencies among the nodes, i.e., the existing network structure affects the creation of an edge, and thus the network links are not independent. In addition, traditional regression analysis is limited to node-pair relationships and cannot analyze how more complex network structures affect network evolution. Existing methods take less account of the time dependence of the network, i.e., the influence of the previous network topology on the current network topology, and cannot accurately reveal the network evolution path. Therefore, we use SAOM to gain insight into the reasons why this network presents the pattern as mentioned above characteristics and evolutionary trends and also construct a virtual water trade time-series evolution model based on structural dependence and time dependence to measure the degree of contribution of different network configurations to the topology of the virtual water network.
The above two aspects are the research deficiencies we derived by combing the existing literature. In the section on research methodology, we point out the problems of traditional correlation analysis in our research topic. We remedy these shortcomings employing more applicable methods. Since we have not sorted out these two main directions more clearly in the paper, we have revised the paper based on your comments.
Point 2: Theoretical framework and assumptions
What is the difference between Hypothesis 3a and 3B, and what are implications of such a difference?
Response 2: Thank you for your valuable comments, we have modified the term. Hypothesis 3B builds on Hypothesis 3a with some additional details. In order to avoid misleading the reader, we have revised both hypotheses so that we can more clearly express the implications of this hypothesis. We remove Hypothesis 3b, which expresses more clearly the verification of the theory of unequal exchange of production, i.e., countries that establish virtual water trade relations with rich countries tend to become net virtual water exporters, with large virtual water resources flowing to developed countries.
Hypothesis 3: Virtual net water exporters tend to increase their export links with rich countries.
Point 3: Materials and Methods
Authors need to describe more details in data sources. I believe present information is inadequate to inform what information, where and how you obtain, especially the information on the virtual water consumption and exchange among trade partners. Additionally,authors have to provide 62 countries and regions and 35 industrial sectors as an appendix.
In table 2, there are both cities and countries in different connections, however I believe they have different roles in global trade for cities and countries, so under what the circumstances at the centre of these trade connections are for cities or countries?
Response 3: Thank you for your valuable comments. We have modified the term. First of all, for the data, we have added more data details based on your suggestions and added details of 62 countries and regions and 35 industrial sectors in the appendix section. In addition, the information on virtual water consumption and the exchange of information between trading partners is a large amount of data, so we compressed and packaged the detailed data files and sent them to the editorial office together. The specific data details have been revised as follows.
The Environmental Extended Input-Output Table is an improvement of the Multi-Regional Inter-Country Input-Output (MRIO) that considers environmental impacts. It is mainly used to describe the direct water consumption by the production process, and the indirect water consumption triggered to meet the final demand in other areas by constructing water satellite use accounts to be integrated with the existing MRIO database. The most common extension is the addition of energy consumption and energy use or energy flows to the traditional MRIO. In order to get the latest global virtual water trade volumes as accurately as possible, two types of data are combined and transformed: value-physical input-output tables and water consumption by sector. The data sources are as follows.
Multi-Regional Inter-Country Input-Output (MRIO) data provide relevant quantitative information at sufficient granularity to describe and analyze production and trade activities. To meet certain information and analysis needs related to Asia and the Pacific, the Asian Development Bank (ADB) added more details on Asian economies to the World Input-Output Database (WIOD), resulting in the ADB Multi-Regional Input-Output Database (ADB-MRIO) (https://data.adb.org/). In the ADB-MRIO, 62 countries and regions and the Rest of the World (RoW) are included between 35 industrial sectors in input-output relationships (Appendix A). In this paper, trade data are taken for three time periods, 2000, 2010, and 2015, and the 35 sectoral trade values are combined and aggregated to the national level. Virtual water use data are collected from the EORA database (https://worldmrio.com/), which includes 35 environmental indicators, including air pollution, energy use, greenhouse gas emissions, water use, and land occupation. In the EORA database, all environmental data are systematically assigned to different sectors in different countries. The database contains green, blue, and gray water usage for 190 countries (regions) from 1990 to 2015. In this paper, the water use data of 62 countries/regions were selected, and the data of green water, blue water, and gray water were integrated to obtain the country's water use in different years. This water use data is used to convert water resources expansion input-output data.
In this paper, referring to the research results of related scholars, the global VWT network is used as the explanatory variable, and eight indicators are selected as explanatory variables from 4 dimensions: economic, social, water resources, and geography (see Table 1). The data of the indicators were selected in 2014 as the time cross-section, and the data sources include World Development Indicators (WDI) (https://databank.worldbank.org/home) and the Center for International Prospective Studies (CEPII) (http://cepii.fr/).
Also, we are very sorry for the error in Table 2. The study in this paper focused on the national level and did not analyze the role of cities in the network, and we have changed all cities to countries. Thank you very much for your suggestion.
Point 4: Results
The organization for subsection 4.2. Analysis on the driving factors of VWT is not somewhat logically explicit, please reorganize this subsection in the way that explicitly accept or reject your hypotheses.
Response 4: Thank you for your valuable comments, we have reorganized this subsection in a way that explicitly accepts or rejects the assumptions. You can see the new, clearer formulation in the revised version.
Point 5: Discussion
Authors should more explain what accepting or denying the hypotheses implies.
Response 5: Thank you for your valuable comments. We have followed your suggestion to analyze the assumptions in depth by elaborating them in more detail in the Discussion section. You can see a more detailed analysis in the revised version.
Point 6: Finally, there are some sentences difficult to understand, which need authors to thoroughly edit them.
Response 6: Thank you for your valuable comments. We are very sorry for some problems with the language presentation, and we have touched up the whole article.
Thank you very much!
